# Rice Bran Phenolic Compounds Regulate Genes Associated with Antioxidant and Anti-Inflammatory Activity in Human Umbilical Vein Endothelial Cells with Induced Oxidative Stress

**DOI:** 10.3390/ijms20194715

**Published:** 2019-09-23

**Authors:** Nancy Saji, Nidhish Francis, Christopher L. Blanchard, Lachlan J. Schwarz, Abishek B. Santhakumar

**Affiliations:** 1Australian Research Council (ARC) Industrial Transformation Training Centre (ITTC) for Functional Grains, Graham Centre for Agricultural Innovation, Charles Sturt University, Wagga Wagga, New South Wales 2650, Australia; nsaji@csu.edu.au (N.S.); nfrancis@csu.edu.au (N.F.); CBlanchard@csu.edu.au (C.L.B.); lschwarz@csu.edu.au (L.J.S.); 2School of Biomedical Sciences, Charles Sturt University, Locked Bag 588, Wagga Wagga, New South Wales 2678, Australia; 3School of Animal and Veterinary Sciences, Charles Sturt University, Locked Bag 588, Wagga Wagga, New South Wales 2678, Australia; 4School of Agricultural and Wine Sciences, Charles Sturt University, Locked Bag 588, Wagga Wagga, New South Wales 2678, Australia

**Keywords:** rice bran, polyphenols, gene expression, endothelial function

## Abstract

Oxidative stress, inflammation and endothelial dysfunction are associated with the development of cardiovascular and metabolic diseases. Phenolic extracts derived from rice bran (RB) are recognised to have antioxidant and anti-inflammatory potential. However, the underlying mechanisms remain unknown. Therefore, this study aimed to evaluate the ability of RB-derived phenolic extracts to modulate genes associated with antioxidant and anti-inflammatory pathways in human umbilical vein endothelial cells (HUVECs) under induced oxidative stress conditions. HUVECs under oxidative stress were treated with varying concentrations of RB phenolic extracts (25–250 µg/mL). Using quantitative real-time polymerase chain reaction, the expression of candidate genes that regulate antioxidant and anti-inflammatory pathways were determined. This included nuclear factor erythroid 2-related factor 2 (*Nrf2*), nicotinamide adenine dinucleotide phosphate: quinone oxidoreductase 1 (*NQO1*), heme oxygenase 1 (*HO1*), nicotinamide adenine dinucleotide phosphate oxidase 4 (*NOX4*), intercellular adhesion molecule 1 (*ICAM1*), endothelial nitric oxide synthase (*eNOS*), ectonucleoside triphosphate diphosphohydrolase 1 (*CD39*) and ecto-5′-nucleotidase (*CD73*). Phenolic extracts derived from RB down-regulated the expression of four genes, *ICAM1*, *CD39*, *CD73* and *NOX4* and up-regulated the expression of another four genes, *Nrf2*, *NQO1*, *HO1* and *eNOS*, indicating an antioxidant/ anti-inflammatory effect for RB against endothelial dysfunction.

## 1. Introduction

Endothelial dysfunction has been identified as a primary contributor to the progression of vascular disorders such as cardiovascular disease (CVD) and stroke [1]. The endothelium maintains vascular haemostasis and exerts anti-coagulant, anti-platelet and fibrinolytic properties [1]. However, oxidative stress resulting from an imbalance between pro-oxidants and antioxidants leads to an increase in the production of reactive oxygen species (ROS) in vascular tissue, consequently leading to the pathogenesis of CVD [2]. Several genes including nuclear factor erythroid 2-related factor 2 (*Nrf2*), nicotinamide adenine dinucleotide phosphate: quinone oxidoreductase 1 (*NQO1*), heme oxygenase 1 (*HO1*), nicotinamide adenine dinucleotide phosphate oxidase 4 (*NOX4*), intercellular adhesion molecule 1 (*ICAM1*), endothelial nitric oxide synthase (*eNOS*), ectonucleoside triphosphate diphosphohydrolase 1 (*CD39*) and ecto-5′-nucleotidase (*CD73*) regulate the production of ROS within the vasculature.

*Nrf2* is a member of the basic leucine zipper transcription factor family that controls the expression of several genes, including *NQO1* and *HO1* [2]. *NQO1*, a cytosolic flavoenzyme, is expressed in tissues including epithelial, vascular endothelial and adipocytes [3], and *HO1* is an enzyme involved in heme catabolism resulting in biliverdin, carbon monoxide and ferrous iron production [4]. Reduced expression and activity of cellular *Nrf2*, *NQO1* or *HO1* results in a failure of antioxidant and cytoprotective enzymes to regulate appropriately under oxidative stress conditions and is therefore associated with increased risk of CVD [3,4]. *eNOS*, primarily found in vascular endothelial cells, is involved in regulating vascular tone and blood clotting by generating protective nitric oxide (NO) molecules in the vasculature [5], thereby preventing the increased ROS production in endothelial cells.

*NOX4*, present in pulmonary arteries as a result of oxidative stress, can result in direct injury to cells, modulate various signalling cascades and regulate transcription factors [6]. *NOX4* is thought to play a role in the regulation of cell growth or cell survival in endothelial cells, suggesting that *NOX4* may play an essential role in the formation of atherosclerosis [6]. Furthermore, soluble cell adhesion molecules such as *ICAM1* have been demonstrated to be responsible for the adhesion of circulating leucocytes to sites of inflammation and their accumulation in arterial walls, subsequently leading to progression of atherosclerosis [7]. During inflammation, purinergic mediators such as adenosine triphosphate (ATP) and adenosine diphosphate (ADP) initiate a series of pro-inflammatory responses [8]. Immunoregulatory enzymes such as *CD39* and *CD73* play a key role in regulating the duration, magnitude and chemical nature of purinergic signals delivered to cells, resulting in adenosine production [8]. The subsequent cascade of events ultimately shifts the pro-thrombotic ATP/ADP-rich environment to an anti-thrombotic, adenosine-rich environment [8]. 

Research into functional foods has considerably increased over the last decade. Consumption of functional foods to reduce the incidence of chronic health disorders has progressively resonated with health-conscious consumers. Several studies, both *in vitro* and *in vivo*, have provided evidence for the protective role of phytochemical compounds as part of a healthy diet, resulting in heart and vasculature protection against oxidative stress [9,10]. In particular, phenolic compounds derived from cereals, fruits and beverages are recognised to have anti-thrombotic activity and improve endothelial function [1]. A recent study conducted by our group has reported that whole grain coloured rice polyphenols may potentially target oxidative stress and inflammatory pathways associated with endothelial dysfunction [7]. 

Rice bran (RB), a by-product of the rice milling process, is recognised to contain a wide range of bioactive chemicals including a combination of polyphenols, γ-oryzanol and tocopherols, which are responsible for most of the antioxidant capacity observed [11]. However, the underlying signalling pathways that contribute to the potential antioxidant and anti-inflammatory properties of RB remains unclear. Therefore, this study aimed to determine the effect of RB phenolic extracts on regulating the expression of antioxidant (*Nrf2*, *NQO1*, *HO1* and *NOX4*) and anti-inflammatory genes (*ICAM1*, *eNOS*, *CD39* and *CD73*) in human umbilical vein endothelial cells (HUVECs) under simulated oxidative stress conditions.

## 2. Results 

### 2.1. Cytotoxicity of RB Phenolic Extracts on HUVECs

The cell viability of HUVECs post 2 h exposure (Figure 1) to different concentrations of RB phenolic extracts has determined the optimal, non-toxic concentrations to be in the range of 25–250 µg/mL. 

### 2.2. Effect of RB Phenolic Extracts on Antioxidant Genes under Oxidative Stress Conditions

The influence of RB phenolic extracts on antioxidant genes in HUVECs (Figure 2) under oxidative stress conditions displayed a significant increase (*p* < 0.05) in the expression of *Nrf2* and *NQO1* genes was observed with pre-treatment at 100 and 250 µg/mL of RB phenolic extracts when compared to the H_2_O_2_ only treated group. The expression of *HO1* gene was significantly increased when pre-treated with 250 µg/mL of RB phenolic extracts. In the *NOX4* gene, a significant reduction in expression (*p* < 0.001) was observed at the highest RB phenolic concentration of 250 µg/mL.

### 2.3. Effect of RB Phenolic Extracts on Anti-Inflammatory Genes under Oxidative Stress Conditions

The influence of RB phenolic extracts on anti-inflammatory genes, *ICAM1*, *eNOS*, *CD39* and *CD73* in HUVECs (Figure 3) under oxidative stress conditions displayed a significant reduction (*p* < 0.0001) in the expression of *ICAM1* and *CD73* across all RB phenolic treatments (25–250 µg/mL). The expression of eNOS gene was significantly increased when pre-treated with 250 µg/mL of RB phenolic extracts. In *CD39* gene, a significant reduction (*p* < 0.01) was observed at 250 µg/mL RB concentration. 

## 3. Discussion 

Endothelial dysfunction is a hallmark of CVD, characterised by high levels of ROS, leading to oxidative stress and inflammation in the vasculature [2]. There are several reports on phenolic compounds that demonstrate protection against cardio-metabolic risk factors such as dyslipidemia, hypertension and glucose metabolism [7,12,13]. However, to the best of the authors’ knowledge, there are no such reports on the cytoprotective effect of RB-phenolic compounds on endothelial function. As such, the present study is, to the best of the authors’ knowledge, the first to detail the effects of RB-phenolic compounds on H_2_O_2_ induced endothelial dysfunction in HUVECs and provides evidence of the potential for RB phenolic compounds to regulate antioxidant and anti-inflammatory signalling pathways. 

RB was found to contain several bioactive chemicals including ferulic acid, *p*-coumaric acid, caffeic acid, vanillic acid, syringic acid, sinapic acid, feruloyl glycoside, shikimic acid, ethyl vanillate, tricin and their isomers with significant antioxidant activity (unpublished data). Previous examinations of phenolic bioavailability have identified that in particular, ferulic acid, one of the most abundant phenolics present in RB, takes 2 h to reach maximal plasma concentration [14]. The cell viability of HUVECs post 2 h H_2_O_2_ exposure (Figure 1) revealed the optimal, non-toxic concentrations of RB extracts to be in the range of 25–250 µg/mL, as such, these concentrations were employed throughout the study. 

### 3.1. Effect on Antioxidant Genes under Oxidative Stress Conditions

Results from this study have demonstrated the antioxidant potential of RB-derived phenolic compounds via up-regulation of *Nrf2*, *NQO1*, and *HO1* expression and down-regulation of *NOX4* expression (Figure 2). *Nrf2* is a transcription factor known to play a significant role in regulating several antioxidant and cytoprotective genes via activation of promoters containing the antioxidant response element [4]. Normally, activation of *Nrf2* up-regulates the production of detoxifying enzymes such as *NQO1* and *HO1* [9]. *HO1* enzyme is usually involved in heme catabolism and is recognised to exert antioxidant affects by nullifying intracellular ROS. *NQO1* enzyme exerts cardioprotective action against oxidative damage by preventing free radical formation from quinone derivatives by transforming the quinone to the redox stable hydroquinone [10]. Therefore, the modulation of *Nrf2* and its downstream antioxidant signalling molecules such as *NQO1* and *HO1* may form the basis for a therapeutic strategy for various chronic diseases. Exposure to cytotoxic ROS has been shown to result in apoptosis of endothelial cells accompanied by reduced transcriptional activity of *Nrf2* [15]. This study demonstrated that phenolic compounds derived from RB have a concentration-dependent increase on *Nrf2*, *NQO1* and *HO1* expression in H_2_O_2_ induced HUVECs (Figure 2). This finding is consistent with literature that demonstrated the effect of other plant polyphenols in modulating *Nrf2* and associated enzyme expression. Patel and Maru [16] confirmed that polyphenols derived from black tea induced *Nrf2* mediated antioxidant responsive element binding in mouse liver and lungs. Additionally, Ungvari et al. [9] have shown that resveratrol provided endothelial protection both *in vitro* and *in vivo* by the activation of *Nrf2*. 

*NOX* isoforms are expressed in several vascular tissues and particularly in HUVECs. The expression level of *NOX4* is recognised to be 100-fold higher than other isoforms, suggesting that *NOX4* is the major source of ROS in the endothelial vasculature [17]. Within resting cells, *NOX* isoforms remain dormant; however, under stimulation via H_2_O_2_, *NOX4* becomes activated, releasing large amounts of superoxides resulting in oxidative stress. Therefore, increased *NOX4* expression is recognised to be associated with early progression of atherosclerotic plaque [17]. This study demonstrated that under H_2_O_2_ induced oxidative stress, *NOX4* expression is significantly up-regulated compared to the DMSO control. However, after treatment with RB phenolic extracts, at 250 µg/mL concentration, the expression of *NOX4* is restored to its original state as observed in DMSO control. Similar findings have been previously reported where they examined the effect of the olive oil phenolic fraction on the angiogenic responses in HUVECs. They also observed that the olive oil extract was able to lower vascular endothelial growth factor-induced angiogenic responses by modulating the expression of several genes, including *NOX4* [13].

### 3.2. Effect on Anti-Inflammatory Genes under Oxidative Stress Conditions

The results from this study demonstrated the ability of RB-derived phenolic compounds to down-regulate the pro-inflammatory genes, *ICAM1*, *eNOS*, *CD39* and *CD73* (Figure 3), thereby indicating potential for anti-inflammatory activity respectively and reducing the impact of ROS. *ICAM1* is a pro-inflammatory cytokine that facilitates endothelial adhesion of circulating leukocytes and is reported to increase under oxidative stress conditions [18]. The presence and increased expression of *ICAM1* in atherosclerotic plaques and lesion-prone hyper-cholesterolemic animals have been previously reported [19]. Therefore, lowering the *ICAM1* expression may have therapeutic benefits against oxidative stress induced CVD as it can prevent migration of leukocytes and prevent further progression of inflammation leading to the formation of fatty streaks in the endothelium [18]. Under H_2_O_2_ induced oxidative stress conditions, treatment with RB phenolic extract resulted in significant down-regulation of *ICAM1* in HUVECs, thereby reducing the inflammatory status of the cells (Figure 3). A similar effect was observed in the study conducted by Callcott et al. [7], where pre-treatment with Reiziq whole grain rice polyphenol extracts significantly reduced *ICAM1* expression (50–250 µg/mL) in HUVECs under oxidative stress. Therefore, the results from this study indicate that the cardioprotective compounds reside mainly in the RB layer as the samples tested were also from Australian grown, Reiziq variety.

*eNOS* enzyme synthesises NO, a potent vasodilator that is identified to play an essential role in protecting against thrombosis and atherogenesis [5]. NO transfers electrons from NADPH through the flavins in the carboxy-terminal reductase domain to the heme in the amino-terminal oxygenase domain [20]. When the flow of electrons through *eNOS* is disturbed, it results in increased ROS production in the vascular endothelial cells. Therefore, a significant decrease in the expression and activity of *eNOS* results in acceleration of atherosclerosis [21]. The present study demonstrates that the expression of *eNOS* gradually increased with RB phenolic extracts, particularly at 250 µg/mL under oxidative stress conditions, suggesting that phenolic compounds in RB ameliorate the oxidative stress in vascular endothelial cells. This is similar to the findings by Madeira et al. [22] where they report that polyphenolic compounds from grape skin extract and red wine respectively induced *eNOS* activation, providing beneficial effects to the cardiovascular system. 

#### Immunomodulatory Genes

Extracellular purinergic mediated cell activation via *CD39* and *CD73* expression of immunomodulatory molecules is recognised to have significant implication in thrombosis, inflammatory response, tissue re-modelling and repair during vascular injury [23]. *CD39* hydrolyses phosphate groups by converting ATP to AMP, which is then dephosphorylated into adenosine by *CD73* [8]. ATP release is a warning signal released by damaged cells that act as an immunostimulatory signal, whereas adenosine is an immunoregulatory signal which results in an anti-inflammatory, anti-thrombotic and vasodilatory response [24]. Consequently, the balance between ATP and adenosine concentration is crucial in immune homeostasis, and therefore, *CD39* and *CD73* expression is finely regulated for appropriate immune response regulation [25]. Polyphenols derived from red wine have been shown to inhibit platelet aggregation and increase *CD39* activity in rat platelets *in vitro* [12]. Another study conducted on wild-type mice has demonstrated that ablation of *CD73* has minimal effect on *in vivo* thrombosis, however, increased *CD39* expression is recognised to attenuate *in vivo* arterial thrombosis [26]. In this study, treatment with RB phenolic extracts reduced the expression of *CD39* and *CD73*. These results are in agreement with Melzig [27], who stated that flavonols such as quercetin, myricetin and kaempferol are known to inhibit the activity of the adenosine deaminase in aortic endothelial cells. It is important to note that the treatment with RB-derived phenolic compounds prevented the increase in oxidative stress caused by H_2_O_2_ exposure (Figure 3). These results support the suggestion that treatment with RB extract was able to avoid the changes caused by H_2_O_2_ exposure, maintaining balanced levels of ATP and adenosine molecules that exhibit potent anti-inflammatory and immunosuppressive actions. Thus, based on our results, we can suggest that the use of RB-derived phenolic compounds is a promising strategy in the treatment of oxidative stress induced by H_2_O_2._


## 4. Materials and Methods 

### 4.1. Reagents 

All chemicals and reagents used in this study were purchased from Promega Corporation (Madison, WI, USA), Bio-Rad (Hercules, CA, USA) or Sigma-Aldrich (St. Louis, MO, USA).

### 4.2. Rice Bran Derived Phenolic Extract Preparation

Commercially stabilised RB (drum-dried), from an Australian grown Reiziq rice variety, was obtained from SunRice milling plant in Leeton, New South Wales and subsequently stored at 4 °C until further analysis. Phenolic compounds were extracted from stabilised RB using acetone/water/acetic acid (70:29.5:0.5, *v*/*v*) mixture as described by Rao et al. [28]. The extract was reconstituted in 10% dimethyl sulfoxide (DMSO) and stored at −20 °C prior to cell culture studies.

### 4.3. Cell Culture Conditions

Experiments were performed with HUVECs purchased from Sigma-Aldrich (St. Louis, MO, USA). HUVECs were maintained in complete endothelial cell growth medium from Cell Applications Inc. (San Diego, CA, USA) at 37 °C in 5% CO_2_ and used before the ninth passage. The cell count for experimental seeding was achieved with Muse^®^ Cell Analyzer from Luminex Corporation (Austin, TX, USA).

### 4.4. Cytotoxicity Assay

Cytotoxicity of RB phenolic extracts was examined using a resazurin red cytotoxicity assay wherein HUVEC cells were seeded into 96-well plates at a density of 5000 cells per well and incubated for 24 h in the complete endothelial cell culture medium. HUVECs were then treated with 200 µL of freshly prepared RB phenolic extract at various concentrations (25, 50, 100, 250, 500, 750 and 1000 µg/mL) for 2, 4, 6 and 8 h time periods. 0.01% DMSO served as a negative control. Subsequently, 200 µL of resazurin red solution (14 mg/L) was added to each well and incubated for an additional 4 h at 37 °C in 5% CO_2_. The absorbance was measured on a microplate reader (FLUOstar Omega microplate reader, BMG Labtech, Offenburg, Germany) at 570 and 600 nm against a resazurin red blank. The percentage of cell viability was calculated using the equation as described by [7]. 

### 4.5. Experimental Design and Induction of Oxidative Stress 

HUVECs were seeded at a density of 300,000 cells per well into 6-well plates and incubated for 24 h. After overnight incubation, cells were treated with RB phenolic extracts (25, 50, 100 and 250 µg/mL) for 2 h. The cells were then washed with pre-warmed phosphate buffered saline (PBS) and treated with a 200 µM concentration of hydrogen peroxide (H_2_O_2_) for 1 h. All experiments were undertaken in triplicates.

### 4.6. Ribonucleic Acid (RNA) Extraction 

The total RNA was isolated using the SV Total RNA Isolation System (Promega, Madison, WI, USA) according to the manufacturer’s instructions. Briefly, cells were washed with ice cold PBS, scraped with 175 μL of RNA lysis buffer containing β-mercaptoethanol. Subsequently, 350 μL of RNA dilution buffer was added and incubated at 70 °C for 3 min. After centrifugation, the supernatant was mixed with 200 μL of 95% ethanol, placed into a spin basket assembly and DNAse digested. The columns were then washed twice with 600 μL RNA solution. RNA was eluted in nuclease-free water and stored at −80 °C until further use. 

### 4.7. Complementary Deoxyribonucleic Acid (cDNA) 

The quality of RNA was determined using a NanoDrop™ 2000c Spectrophotometer from Thermo Fisher Scientific (Waltham, MA, USA). After which, cDNA synthesis was conducted using GoScript™ Reverse Transcriptase (Promega, Madison, WI, USA) as per the manufacturer’s instructions. Briefly, Oligo (dT)_15_ Primer and a reverse transcription reaction mix containing GoScript™ 5× reaction buffer, MgCl_2_ (final concentration 1.5–5.0 mM), PCR nucleotide mix (final concentration 0.5 mM each dNTP), recombinant RNAsin^®^ ribonuclease inhibitor (optional), GoScript™ reverse transcriptase and nuclease-free water was added to each sample and cDNA synthesis was performed. The cycling conditions comprised of annealing at 25 °C for 5 min, extension at 42 °C for 1 h and inactivation of reverse transcriptase for 15 min at 70 °C.

### 4.8. Primer Sequences

Primers used for quantitative real-time polymerase chain reaction (qPCR) examinations are listed in Table 1. All of the qPCR primers were designed using Primer3 software and synthesized by Sigma-Aldrich (St. Louis, MO, USA). The amplification efficiency was determined to be in the range of 90–110% for all the primers prior to commencing qPCR. 

### 4.9. Quantitative Real-Time Polymerase Chain Reaction (qPCR)

Gene expression was conducted in the CFX96 Touch™ Real-Time PCR Detection System (Bio-Rad) using SsoAdvanced™ Universal SYBR^®^ Green Supermix (Bio-Rad) detection according to the manufacturer’s instructions. Briefly, SsoAdvanced™ Universal SYBR^®^ Green Supermix (9 µL) containing forward and reverse primer was added to 96-well PCR plate along with 1 µL of cDNA in each well. *β-actin* served as the reference gene. The cycling conditions comprised of 95 °C for 3 min, 95 °C for 10 s and 58 °C for 30 s repeated for 40 cycles. Melt curve was generated at 65 °C for 5 s and 95 °C for 50 s. The endpoint or cycle threshold (*C*_t_) values were obtained for all genes tested. The mean normalized expression of genes were determined using Q-gene software application as described by Muller et al. [29]. *β-actin* served as the housekeeping gene 

### 4.10. Statistical Analysis 

Statistical analysis was performed using one-way analysis of variance (ANOVA), followed by post-hoc Tukey’s multiple comparisons test using GraphPad Prism 7 software (GraphPad Software Inc., San Diego, CA, USA) at a level of *p* < 0.05. The results are reported as mean ± standard error of mean (SEM).

## 5. Conclusions

In summary, RB-derived phenolic compounds displayed a cytoprotective effect on vascular endothelial cells under oxidative stress. Pre-treatment with RB phenolic extracts resulted in the regulation of antioxidant (*Nrf2*, *NQO1*, *HO1* and *NOX4*) and anti-inflammatory (*ICAM1*, *eNOS*, *CD39* and *CD73*) genes. Since RB contains several bioactive chemicals including *p*-coumaric acid, caffeic acid, vanillic acid, ferulic acid and syringic acid, the effects observed in this study may be due to the synergistic effect of these bioactive compounds. With relatively few human clinical studies on the antioxidant activities of cereals being currently available, information regarding the bioavailability of polyphenolic extracts is limited. Therefore, *in vivo* metabolomic studies are essential to examine the antioxidant and anti-inflammatory potential of cereal grains to essentially utilise functional cereal grain by-products such as RB.

## Figures and Tables

**Figure 1 ijms-20-04715-f001:**
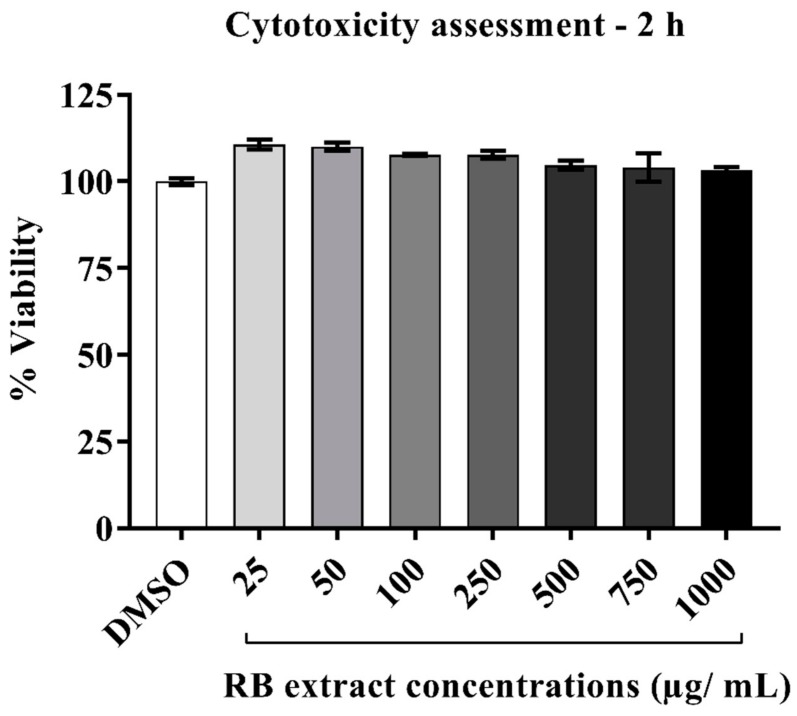
Cytotoxicity results in HUVECs after 2 h post-exposure to different concentrations of RB phenolic extracts. The RB phenolic extracts did not display any cytotoxic effect on the HUVEC cells at any of the concentrations tested (25–1000 µg/mL) (*n* = 3). Data is presented as mean ± SEM. Dimethyl sulfoxide, DMSO; human umbilical vein endothelial cells, HUVEC; rice bran, RB.

**Figure 2 ijms-20-04715-f002:**
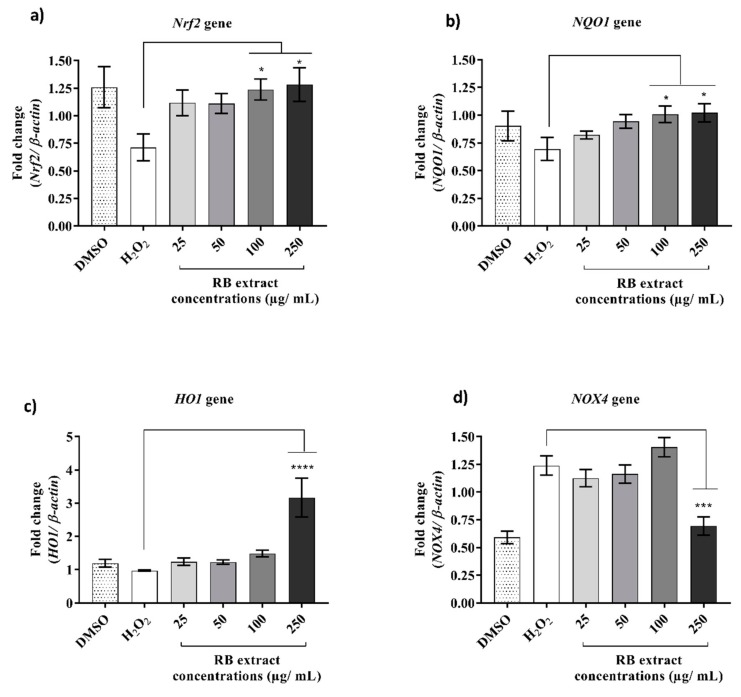
Effect of RB phenolic extracts on antioxidant genes. (**a**) *Nrf2* (**b**) *NQO1* (**c**) *HO1* (**d**) *NOX4* in HUVECs under oxidative stress conditions. A significant increase (*p* < 0.05) in the expression of *Nrf2* and *NQO1* genes was observed with pre-treatment at 100 and 250 µg/mL of RB phenolic extracts when compared to the H_2_O_2_ only treated group. The expression of *HO1* gene was significantly increased when pre-treated with 250 µg/mL of RB phenolic extracts. In the *NOX4* gene, a significant reduction in expression (*p* < 0.001) was observed at the highest RB phenolic concentration of 250 µg/mL. The level of significance is indicated by the asterisks, whereby * *p* < 0.05, *** *p* < 0.001, **** *p* < 0.0001. *n* = 3. Data is presented as mean ± SEM. Dimethyl sulfoxide, DMSO; hydrogen peroxide, H_2_O_2_; human umbilical vein endothelial cells, HUVEC; rice bran, RB; nuclear factor erythroid 2-related factor 2 (*Nrf2*); NADPH: quinone oxidoreductase 1 (*NQO1*); heme oxygenase 1 (*HO1*); NADPH oxidase 4 (*NOX4*).

**Figure 3 ijms-20-04715-f003:**
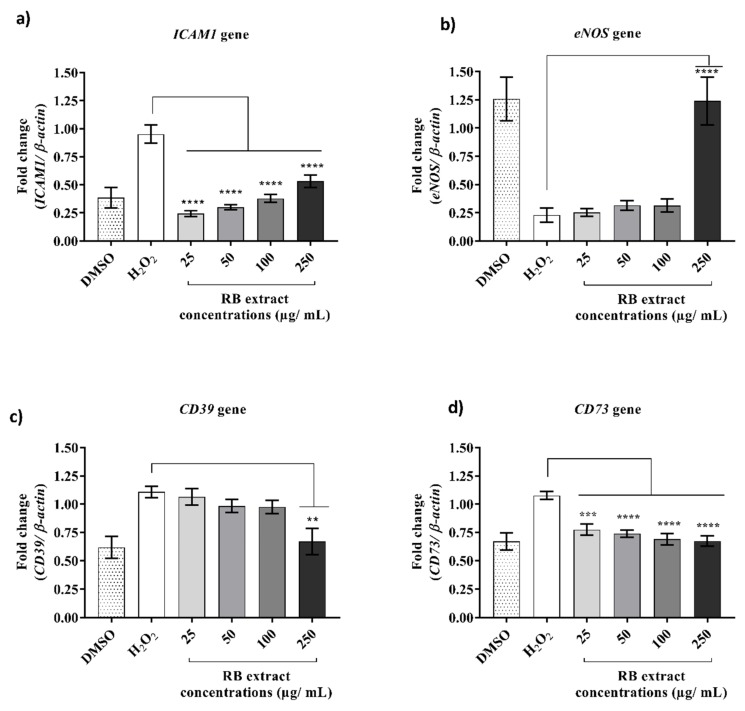
Effect of RB phenolic extracts on anti-inflammatory genes. (**a**) *ICAM1* (**b**) *eNOS* (**c**) *CD39* (**d**) *CD73* in HUVECs under oxidative stress conditions. A significant reduction (*p* < 0.0001) in the expression of *ICAM1* and *CD73* was seen across all RB phenolic treatments (25–250 µg/mL). The expression of *eNOS* gene was significantly increased when pre-treated with 250 µg/mL of RB phenolic extracts. In *CD39* gene, a significant reduction (*p* < 0.01) was observed at 250 µg/mL RB concentration. The level of significance is indicated by the asterisks, whereby ** *p* < 0.01, *** *p* < 0.001, **** *p* < 0.0001. *n* = 3. Data is presented as mean ± SEM. Dimethyl sulfoxide, DMSO; hydrogen peroxide, H_2_O_2_; human umbilical vein endothelial cells, HUVEC; rice bran, RB; intercellular adhesion molecule 1 (*ICAM1*); endothelial nitric oxide synthase (*eNOS*); ectonucleoside triphosphate diphosphohydrolase 1 (*CD39*); ecto-5′-nucleotidase (*CD73*).

**Table 1 ijms-20-04715-t001:** The nucleotide sequences of the PCR primers used to assay gene expression by qPCR.

Gene	Forward Primer	Reverse Primer
*Nrf2*	ATGACAATGAGGTTTCTTCGG	CAATGAAGACTGGGCTCTC
*NQO1*	ACATCACAGGTAAACTGAAGG	TCAGATGGCCTTCTTTATAAGC
*HO1*	AACTCCCTGGAGATGACTC	CTCAAAGAGCTGGATGTTGAG
*eNOS*	GTTACCAGCTAGCCAAAGTC	TCTGCTCATTCTCCAGGTG
*NOX4*	TATCCAGTCCTTCCGTTGG	CCAATTATCTTCTGTATCCCATCTG
*ICAM-1*	GATAGCCAACCAATGTGCT	TTCTGGAGTCCAGTACACG
*CD39*	TCAAATGTAGTGTGAAAGGCTC	TACACTCCTCAAAGGCTCTG
*CD73*	CATTCCTGAAGATCCAAGCA	AGGAGCCATCCAGATAGAC
*β-actin*	GAAGATCAAGATCATTGCTCCTC	ATCCACATCTGCTGGAAGG

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
