# Peer review of "Rice Bran Phenolic Compounds Regulate Genes Associated with Antioxidant and Anti-Inflammatory Activity in Human Umbilical Vein Endothelial Cells with Induced Oxidative Stress"

_ijms, 2019, doi:10.3390/ijms20194715_

Round 1
Reviewer 1 Report
The authors described rice bran phenolic compounds regulating genes associated with antioxidant and anti-inflammatory activity in human umbilical vein endothelial cells. The experimental results were interesting for the researchers in the related field. The manuscript may be worthy for publication in this journal after revisions/additions outlined below:
1) The formatting for reference citation should be followed in author guidance of IJMS.
2) The numbering of subtitle should be corrected.
Materials and Methods -> 2. Materials and Methods2.6. Complementary deoxyribonucleic acid (cDNA) -> 2.7. Complementary deoxyribonucleic acid (cDNA). Followings should be also corrected.
Results -> 3. Results Discussion -> 4. Discussion3) The chemical profiling of RB phenolic extract should be analyzed by HPLC or LC/MS.
4) Figures 2 and 3 can be changed to much higher resolution file.
5) Authors should have the section of Conclusion. There is no conclusion.
6) Reference list should be followed by the reference style of IJMS.
Author Response
The formatting for reference citation should be followed in author guidance of IJMS.All the reference formatting has now been corrected.
The numbering of subtitle should be corrected.Materials and Methods -> 2. Materials and Methods
2.6. Complementary deoxyribonucleic acid (cDNA) -> 2.7. Complementary deoxyribonucleic acid (cDNA).
Results -> 3. Results
Discussion -> 4. Discussion
All the numbering of subtitles has now been corrected.
The chemical profiling of RB phenolic extract should be analyzed by HPLC or LC/MS.The chemical profiling of the rice bran was performed using uHPLC and LC-QTOF-MS. It was observed that ferulic acid, p-coumaric acid, caffeic acid, vanillic acid, syringic acid, sinapic acid, feruloyl glycoside, shikimic acid, ethyl vanillate, tricin, and their isomers were the predominant antioxidant compounds (this has been included in the discussion section lines 223-224). Since considerable characterisation of phenolic compounds evaluating the impact of stabilisation process on rice bran in modulating phenolic profile was performed, a standalone original article is currently under review by another journal.
Figures 2 and 3 can be changed to much higher resolution file.Higher resolution images has now been added.
Authors should have the section of Conclusion. There is no conclusion.A conclusion section has now been added in the manuscript (Lines 313 to 323)
Reference list should be followed by the reference style of IJMS.Reference list is now changed to follow IJMS style.
Reviewer 2 Report
Abstract:
Mentioned the no. of genes are up-regulated and down-regulated along with major genes.
Introduction:
Rice‐derived phenolic compounds already study and published for having antioxidant and anti‐inflammatory potential by several workers How this study is new particularly for antioxidant and anti‐inflammatory potential.
Results: There is no significant cytotoxicity seen in the graph at different concentration. What’s the reason?
No discussion about these results in the discussion section.
In all the manuscript references not as per the journal format.
Follow the journal instruction for the references in the manuscript

Author Response
Abstract: Mentioned the no. of genes are up-regulated and down-regulated along with major genes.The number of genes up-regulated and down-regulated has now been added in the abstract.
Line 38 – 41 now reads:
Phenolic extracts derived from RB down-regulated the expression of four genes, ICAM1, CD39, CD73, and NOX4 and up-regulated the expression of another four genes, Nrf2, NQO1, HO1 and eNOS indicating an antioxidant/ anti-inflammatory effect for RB against endothelial dysfunction.
Introduction: Rice‐derived phenolic compounds already study and published for having antioxidant and anti‐inflammatory potential by several workers. How this study is new particularly for antioxidant and anti‐inflammatory potential.Although there have been previous studies that investigates rice‐derived phenolic compounds have antioxidant and anti‐inflammatory potential, evaluation of gene expression pathways specifically targeting molecular mechanistic pathways on inflammation and oxidative/free radical damage upon rice bran treatment has not been investigated. A coherent brief of the various pathways rice bran has impacted are presented in detail in the discussion section.
Results: There is no significant cytotoxicity seen in the graph at different concentration. What’s the reason? No discussion about these results in the discussion section.The results demonstrate that rice bran derived polyphenols (upto 1000 ug/ml) do not have cytotoxic effects (at 2, 4, 6 or 8 hours) on the cell line tested. A 2 hour time point was chosen for the remainder of the experiment based on phenolic bioavailability. For example, ferulic acid, one of the most abundant phenolic present in RB, takes 2 h to reach maximal plasma concentration (line 225 -227).
In all the manuscript references not as per the journal format. Follow the journal instruction for the references in the manuscript.References have now been corrected to follow the journal format.
Round 2
Reviewer 1 Report
All comments were well addressed.